# Phytogenic Synthesis of Cuprous and Cupric Oxide Nanoparticles Using *Black jack* Leaf Extract: Antibacterial Effects and Their Computational Docking Insights

**DOI:** 10.3390/antibiotics13111088

**Published:** 2024-11-14

**Authors:** Sutha Paramasivam, Sathishkumar Chidambaram, Palanisamy Karumalaiyan, Gurunathan Velayutham, Muthusamy Chinnasamy, Ramar Pitchaipillai, K. J. Senthil Kumar

**Affiliations:** 1PG and Research Department of Chemistry, Government Arts College (Affiliated to Bharathidasan University), Ariyalur 621713, Tamil Nadu, India; psudha233@gmail.com; 2Nextgen Academic Research, Perambalur 621212, Tamil Nadu, India; csathishkumar@nmc.ac.in; 3Research Department of Chemistry, School of Science and Humanities, Dhanalakshmi Srinivasan University, Perambalur 621212, Tamil Nadu, India; palanisamyk.set@dsuniversity.edu.in; 4Research Department of Chemistry, Bishop Heber College (Affiliated to Bharathidasan University), Tiruchirappalli 620017, Tamil Nadu, India; gurunathan.ch@bhc.edu.in; 5Department of Biotechnology, Srinivasan College of Arts and Science (Affiliated to Bharathidasan University), Perambalur 621212, Tamil Nadu, India; biotech@scas.ac.in; 6Center for General Education, National Chung Hsing University, Taichung 402, Taiwan

**Keywords:** bactericide, *biogenesis*, Cu_2_O, CuO, *in-silico* docking, nanodrugs

## Abstract

**Background:** Green synthesized nanoparticles (NPs) have gained increasing popularity in recent times due to their broad spectrum of antimicrobial properties. This study aimed to develop a phytofabrication approach for producing cuprous (Cu_2_O) and cupric oxide (CuO) NPs using a simple, non-hazardous process and to examine their antimicrobial properties. **Methods:** The synthesis employed *Bidens pilosa* plant extract as a natural reducing and stabilizing agent, alongside copper chloride dihydrate as the precursor. The biosynthesized NPs were characterized through various techniques, including X-ray diffraction (XRD), transmission electron microscopy (TEM), Fourier-transform infrared (FT-IR) spectroscopy, ultraviolet–visible (UV-Vis) spectroscopy, scanning electron microscopy (SEM), and energy-dispersive X-ray spectroscopy (EDS). **Results:** XRD analysis confirmed that the synthesized CuO and Cu_2_O NPs exhibited a high degree of crystallinity, with crystal structures corresponding to monoclinic and face-centered cubic systems. SEM images revealed that the NPs displayed distinct spherical and sponge-like morphologies. EDS analysis further validated the purity of the synthesized CuO NPs. The antimicrobial activity of the CuO and Cu_2_O NPs was tested against various pathogenic bacterial strains, including *Staphylococcus aureus*, *Pseudomonas aeruginosa*, *Escherichia coli*, and *Bacillus cereus*, with the minimum inhibitory concentration (MIC) used to gauge their effectiveness. **Conclusions:** The results showed that the phytosynthesized NPs had promising antibacterial properties, particularly the Cu_2_O NPs, which, with a larger crystal size of 68.19 nm, demonstrated significant inhibitory effects across all tested bacterial species. These findings suggest the potential of CuO and Cu_2_O NPs as effective antimicrobial agents produced via green synthesis.

## 1. Introduction

The occurrence of infectious illnesses poses a significant global danger to public health, particularly due to the rise in antibiotic-resistant pathogenic microbes. The types of bacteria that are classified as Gram-positive or Gram-negative are recognized as an important threat to the public’s health. Consequently, antibacterial agents have been employed to manage illnesses that arise in both community and hospital environments [1]. Because of the rise in microbes that are impermeable to numerous antibiotics, there is an increasing desire to create novel antibiotics using inorganic substances as a replacement for conventional organic compounds. This is because organic substances have restricted uses owing to their poor resistance to heat, extreme susceptibility to decomposition, and shorter lifespan [2,3]. In recent times, increased attention has been paid to metal oxide nanoparticles (NPs) comprising copper oxide (CuO) NPs owing to their distinct physical, biological, and optical characteristics [4,5,6]. 

CuO NPs have been taken into consideration because of their important significance for the ecological treatment and biomedical sectors [7]. CuO NPs can be fashioned over a variety of chemical and physical techniques, together with sol–gel synthesis, thermal degradation, sonochemical approaches, microwave irradiation, solution plasma, chemical precipitation, and electrochemical reduction [8,9,10,11,12,13,14,15,16]. Despite the simplicity and great efficiency of these nanoparticle production processes, the utilization of harmful compounds as reducers constitutes a significant issue of concern. The utilization of nonpolar solvents in specific techniques enhances the toxicity of CuO NPs during their formation. Hence, employing biological materials for the production of nanoparticles could offer a beneficial, environmentally friendly, and ecologically sound method [17]. 

Microbes like bacteria, fungi, algae, and plant sources were proposed as sustainable replacements to chemical processes for synthesizing nanoparticles. These biological approaches are cost-effective, energy-efficient, and not harmful [18]. The utilization of plant sources for the production of nanoparticles has several advantages compared to alternative ecologically sound biological techniques since it removes the need for the complex task of continuing cell culture [19]. In addition, employing botanicals for the formation of nanoparticles has the benefit of their easy accessibility, safe handling, and wide range of phytochemicals that can assist in the reduction process [20,21,22]. 

CuO nanoparticles were synthesized from different extracts of plants, including *Carica papaya*, *Gloriosa superba*, *Catha edulis*, *Malva sylvestris*, *Calotropis gigantean*, *Terminalia arjuna*, *Phyllanthus amarus*, and *Acalypha indica*, as documented in earlier research [23,24,25,26,27,28,29]. The antimicrobial effects of these nanoparticles were also documented. This study was conducted to look into the utilization of *Bidens pilosa* leaf extract as a regulating and reducing substance for synthesizing CuO and Cu_2_O NPs. Additionally, it intended to assess the antibacterial action of the phytofabricated CuO and Cu_2_O NPs towards specific infectious microbes employing the agar disk diffusion technique. *Bidens pilosa* leaf extract is also widely consumed as a psychostimulant medication similar to amphetamines. Moreover, this study provides an opportunity for future investigations into the utilization of numerous Indian-origin herbs for the production of nanoparticles, which can be applied in diverse fields.

## 2. Results and Discussion

### 2.1. UV–Visible Studies

The UV–visible absorption peak, which occurs between 280 and 400 nm, indicates the development of CuO NPs. In this research, the absorbance that appears at 365 nm corresponds to the specific surface plasmon resonance (SPR) peak for CuO NPs with smaller particle sizes. Figure 1a exhibits the UV–visible spectra of CuO NPs that were manufactured using a more environmentally friendly method. 

The production of Cu_2_O NPs has been verified via UV–visible spectroscopy. UV–visible spectroscopy is a crucial method for determining the creation and durability of nanomaterials in a water-based system. Figure 1b depicts the UV-vis data obtained for Cu_2_O nanoparticles. The change in color is instigated by the surface plasmon vibrations in the Cu_2_O NPs, which are visible at a wavelength of 663 nm, indicating its identification.

### 2.2. FT-IR Studies

The FT-IR spectra are observed within the array of 500–4000 cm^−1^. The presence of a broad peak at 3428.71 cm^−1^ resembles the O-H functional moiety, which is likely owed to the occurrence of a hydroxyl component [30]. The spectral band seen at the wavenumber of 2924.03 cm^−1^ indicates the existence of the C-H component [31]. The occurrence of a carbonyl moiety (C=O) was confirmed by the peaks seen at 1626.66 cm^−1^, indicating the presence CO_2_ [32]. The peak at 1482.20 cm^−1^ represents the aliphatic -CH moiety. The peak at 1368.36 cm^−1^ signifies the bending vibration of -OH moiety. The peak at 1044.71 cm^−1^ describes the presence of a C-O bond. The peak at 601.63 cm^−1^ describes the bending vibration of the -CH group. The presence of Cu-O interactions is confirmed by the peak seen at 516.34 cm^−1^ [33]. The FT-IR spectra of CuO NPs fashioned using a green method are depicted in Figure 2a.

An FT-IR spectroscopy examination was conducted on the biosynthesized Cu_2_O NPs (Figure 2b) to establish the functional moieties involved in their production and stabilization. The produced Cu_2_O NPs were analyzed using FTIR measurements (Figure 2b) to discover potential biomolecules in the *Bidens pilosa* extract that serve as both capping and reducing agents. The band at 3247 cm^−1^ signifies the existence of O-H bonds [18,34,35,36,37]. The peak at 3105 cm^−1^ signifies the presence of the -NH group. The appearance of -CH moiety is represented by the peak at 2972 cm^−1^. The peeks seen at 1461 cm^−1^ and 1648 cm^−1^ are attributed to the symmetric and asymmetric vibrations of carboxylates (COO^–^), accordingly [38,39]. The peak at 1226 cm^−1^ represents the carbonyl moiety of a ketone. The peak at 1094 cm^−1^ describes the presence of a C-O bond. The peak at 766 cm^−1^ describes the bending vibration of the -CH group. The 646 cm^−1^ band is recognized as the vibration of Cu(I)–O in Cu_2_O NPs [40,41,42]. The peaks attributed to C=C and C=O are believed to originate from ambient moisture and CO_2_.

### 2.3. SEM Studies 

An SEM examination was conducted to determine the morphological characteristics of CuO NPs fashioned with an aqueous leaf extract of *Bidens pilosa*. The SEM image depicted in Figure 3a displayed that the CuO NPs had a spherical morphology. The CuO NPs, which were manufactured in a green manner, were distributed as separate particles and exhibited monodispersivity. The aqueous leaf extract of *Bidens pilosa* contains phytonutrients that function as a capping agent. This capping agent prevents the particles from aggregating, resulting in the monodispersivity of CuO NPs.

The research via SEM studies was conducted to govern the morphological characteristics of Cu_2_O NPs generated with a green method with *Bidens pilosa*. The SEM image depicted in Figure 3b specified that the synthesized Cu_2_O NPs had a spherical form. The Cu_2_O nanoparticles, which were manufactured in a green manner, were disseminated as individual particles and exhibited monodispersivity. The aqueous leaf extract of *Bidens pilosa* contains phytonutrients that function as capping agents, preventing the agglomeration of particles and resulting in the monodispersivity of Cu_2_O NPs.

### 2.4. EDX Analysis

The elements present in the produced CuO NPs were verified using EDX examination. The presence of peaks corresponding to oxygen and copper atoms in the EDX spectra provided confirmation that the produced substance consisted of CuO NPs (Figure 4a). The % weight of oxygen and copper atoms was 28.90 and 71.10, respectively. There are no other peaks detected in the spectra other than copper and oxygen, signifying the purity of CuO NPs. The elemental makeup of CuO NPs is reported in Table 1. The structural makeup of the produced Cu_2_O nanoparticles was verified using the EDX technique. The existence of distinct peaks consistent with oxygen and copper in the EDX spectra provided confirmation that the produced material consisted of Cu_2_O NPs (Figure 4b). The weight fraction of oxygen and copper atoms was 33.06% and 66.94% respectively. There are no additional peaks observed in the spectra other than copper and oxygen, signifying the purity of Cu_2_O NPs. Table 1 displays the chemical makeup of Cu_2_O NPs.

### 2.5. TEM Studies

The shape and dimensions of the synthesized nanoparticles were analyzed using TEM assessment. Figure 5a presents a representative TEM micrograph of synthesized CuO NPs, illustrating the formation of spherical particles with a smooth surface. A considerable quantity of uniformly shaped tiny particles was observed in the micrograph. The determined average diameter of CuO nanoparticles is 12 nm. Figure 5b illustrates the TEM image of the synthesized Cu_2_O NPs. The TEM picture of Cu_2_O reveals the uniform distribution of approximately spherical particles. The average dimensions of Cu_2_O particles are around 67 nm. The dimensions of Cu_2_O nanoparticles derived from TEM analyses closely align with those determined from XRD studies.

### 2.6. XRD Studies

Figure 6a displays the XRD spectra of CuO NPs. The peaks seen at 2θ = 32.2°, 35.3°, 38.5°, 45.9°, 48.6°, 53.4°, 58.2°, 61.4°, 66.0°, 67.8°, 72.2°, and 75.0° correspond to the (110), (−111), (111), (112), (202), (020), (202), (113), (310), (220), (311), and (310) planes of the CuO NPs. The peaks closely match the conventional structure for the crystalline monoclinic phase of CuO NPs (JCPDS No. 80-0076). No contaminant peaks were detected. The pronounced peaks reveal the highly crystalline character of the produced NPs. The average size of the crystals can be assessed by applying the Scherer equation to the primary diffracted peak [43,44]: $$D_{\left(hkl\right)}=\frac{k\lambda }{\beta cos\theta }$$

The average crystallite size, *D*_(*hkl*)_, is assessed by the full width at half maximum (FWHM) denoted by *β*, the wavelength of the incident X-ray (*λ* = 0.15405 nm) from a Cuk_α_ source, a shape constant (*k* = 0.89), and the incident angle of the X-ray, *θ*. The mean size of the CuO NPs that were produced was 12.45 nm. The XRD patterns of Cu_2_O NPs were obtained in the angular range of 20 to 100°. Figure 6b displays the XRD spectra of Cu_2_O NPs. The diffractogram of Cu_2_O NPs exhibited two theta peaks at 29.67, 36.56, 42.47, 61.62, 73.83, and 77.69, equivalent to the (1 1 0), (1 1 1), (2 0 0), (2 2 0), (3 1 1), and (2 2 2) Miller indices, accordingly. The two theta peaks of Cu_2_O NPs were aligned with Standard JCPDS-05-0667 (Cu_2_O). The findings indicated that the Cu_2_O NPs exhibited a face-centered cubic (FCC) lattice phase. The mean crystallite size of the produced Cu_2_O NPs was 68.19 nm.

### 2.7. In Vitro Antibacterial Effect 

In vitro antibacterial studies were conducted using CuO and Cu_2_O NPs and compared with the conventional medicine ciprofloxacin towards microbes of *E. coli*, *P. aeruginosa*, *B. cereus*, and *S. aureus*. The agar well diffusion method was employed to examine the MIC of an antibiotic. Table 2 displays the MICs for the aqueous extract of *Bidens pilosa* leaves, as well as the produced CuO and Cu_2_O NPs. The antibacterial efficacy of synthesized Cu_2_O NPs is significantly greater than that of *Bidens pilosa* leaf extract. The MIC values of Cu_2_O and CuO NPs towards *E. coli* are 16 and 18 µg/mL, respectively, which signifies excellent antibacterial action compared to standard ciprofloxacin with an MIC value of 20 µg/mL. The Cu_2_O and CuO NPs displayed a significant antibacterial effect towards *B. cereus* with an MIC of 24 and 26 µg/mL compared to the control medication ciprofloxacin (MIC: 50 µg/mL). The Cu_2_O nanoparticle exhibited a significant antibacterial effect with an MIC of 26 and 20 µg/mL towards *P. aeruginosa* and *S. aureus*, in comparison to conventional ciprofloxacin. The produced CuO and Cu_2_O NPs have a significantly greater antibacterial action towards the microbes *S. aureus* and *E. coli* compared to the control ciprofloxacin. The produced Cu_2_O NPs displayed exceptional antibacterial action towards all the investigated microbial strains. The Cu_2_O NPs show greater antibacterial activity than CuO NPs; although the size of the Cu_2_O NPs is greater than CuO NPs, they show greater activity as shown by previous studies [45]. We strongly report that our studies are in line with other studies in the literature.

### 2.8. Docking Studies 

The docking behavior of the synthesized CuO and Cu_2_O NPs, as well as the control ciprofloxacin, was examined with respect to protein 1KZN using the Autodock Vina program [46]. The synthesized Cu_2_O NPs demonstrated significant inhibiting ability with a binding score of −8.2 kcal/mol associated with the CuO NPs, which had a binding score of −6.4 kcal/mol, and the control ciprofloxacin, which had a binding score of −6.0 kcal/mol, with the 1KZN protein. The synthesized CuO and Cu_2_O NPs do not engage in any hydrogen bond interactions with the 1KZN receptor. Asp73 and Thr165 residues of amino acids were involved in hydrophobic interactions within the CuO:1KZN complex. The residues amino acids, namely Ile90, Val93, and Val118 were involved in hydrophobic interactions in the Cu_2_O:1KZN complex. In contrast, the control ciprofloxacin does not establish any hydrogen bond association with the 1KZN receptor. In addition, the residue amino acids Asn46, Ala47, Glu50, Asp73, Ile78, and Ile90 were involved in hydrophobic interactions. Figure 7 shows the hydrophobic contacts between the residues of amino acids in the 1KZN receptor and CuO (Figure 7a) and Cu_2_O (Figure 7b) NPs and ciprofloxacin (Figure 7c) that were elucidated. The outcomes designate that the artificially produced Cu_2_O NPs demonstrate a substantial capacity for inhibition when compared to CuO NPs and the control substance ciprofloxacin in terms of antibacterial testing. Table 3 shows a summary of the outcomes.

## 3. Materials and Methods

### 3.1. Chemistry 

Sigma Aldrich’s (St. Louis, MO, USA) copper chloride dihydrate (CuCl_2_·2H_2_O) was utilized unprocessed because of its high analytical quality. The local marketplace in Tiruchirappalli, Tamil Nadu, India, provided the fresh *Bidens pilosa* leaves used in this study. Distilled water (DW) was utilized for all the aqueous solutions.

### 3.2. Bidens Pilosa (BP) Extract Preparation 

A weight of 400 g of fresh BP leaves was washed in DW to eliminate any dirt or dust. When the leaves were cleaned, they were sliced into chunks and pulverized through mortar and pestle in the lab. We used distilled water to homogenize the aqueous extract, then strained it through a mesh, a filter cloth, and finally some Whatmann No.1 filter paper (Sigma Aldrich). After filtering, the extract was kept at 4 °C in a freezer until required.

### 3.3. Biogenesis of the Cu_2_O NPs 

A standard synthesis approach involved dissolving 1.7 g of CuCl_2_·2H_2_O in 100 mL of distilled water, and then thoroughly mixing the mixture for each sample. Next, 40 mL of *Bidens pilosa* leaf extract solution was added, stirring vigorously for one hour at room temperature. Then, 10% NaOH solution was slowly introduced to the reaction mixture in order to correct its pH. Throughout this procedure, the solution underwent a progressive transformation, shifting from its original color to a distinct brick-red hue. This transformation signifies the creation of cuprous oxide (Cu_2_O). The subsequent solution was exposed to reflux and then permitted to precipitation. The resulting substance was then filtered and sequentially rinsed with ethanol, distilled water, and acetone. The Cu_2_O NPs were subsequently dried out at a temperature of 120 °C for over 3 h in order to achieve a finely powdered substance.

### 3.4. Phytogenesis of CuO NPs 

A total of 0.05 M of CuCl_2_·2H_2_O solution was generated by combining it in 30 mL of distilled water and agitating it with a magnetic stirrer. In total, 30 mL of plant extract was introduced and stirred continuously, after 10 min of continuous stirring of the precursor solution. After stirring for 20 min, a 10% NaOH solution was slowly introduced to the reaction mixture in order to correct its pH. Finally, the solid obtained was centrifuged at a speed of 6000 rpm, undergoing multiple rinses using water and ethanol. The CuO NPs were subsequently dried out at a temperature of 150 °C for over 4 h. The finalized substance was pulverized into a small particle using a pestle and mortar.

### 3.5. Characterization Studies 

The CuO and Cu_2_O nanoparticles produced via BP were examined with numerous spectroscopic techniques. The UV–visible spectra for synthesized CuO and Cu_2_O were attained by using a V-730 UV–visible spectrophotometer (JASCO, Hachioji, Japan), which conducted measurements at a wavelength of 200–800 nm. The FT-IR spectra were obtained utilizing an FT/IR-6600 spectrometer (CHI 1000C, Bee Cave, TX, USA) at 4000–400 cm^−1^. The XRD investigation was conducted via the X’Pert Pro instrument manufactured by PANalytical (Malvern, UK). FE-SEM imaging, together with EDX, was performed using the FESEM sigma crucial instrument manufactured by Zeiss Microscopy (Jena, Germany).

### 3.6. Antibacterial Studies

The antibacterial properties of BP aqueous leaf extract (1), CuO NPs (2), and Cu_2_O NPs (3) were tested on several bacterial cultures, including *Pseudomonas aeruginosa*, *Escherichia coli*, *Bacillus cereus*, and *Staphylococcus aureus*, that have been cultured on the nutrient’s agar media. The antibacterial behavior was carried out as per the guidelines set out by the Institute of Clinical and Laboratory Standards [47]. An assay to disseminate the disks was used to investigate bacterial immunity to CuO and Cu_2_O NPs. In sterile deionized water with a serial dilution of CuO and Cu_2_O NPs, studies were performed in triplicate (200, 100, 50, 25, and 12.5 µg/mL). The specimens were first incubated at 4 °C for 15 min before being moved to 37 °C overnight. Whenever a zone of inhibition was discovered throughout the well after the incubation period, the width of the zones of inhibition was calculated using a digital vernier caliper [48].

### 3.7. Determination of MIC 

The microbial cultures often used to make 0.5 McFarland were cultured overnight at 37 °C. Every culture was inoculated in aseptic conditions with 1 mL of the specific bacterial culture (around 108 CFU/mL) from a minimum of a 10 mL tube nutritional broth medium. In sterile deionized water, five dilutions of *Bidens pilosa* aqueous leaf extract (1), CuO NPs (2), and Cu_2_O NPs (3) (200, 100, 50, 25, and 12.5 µg/mL) were made, as well as a blank sample (without CuO and Cu_2_O NPs). All isolate evaluations were carried out in triplicate. The implanted tubes were maintained at 37 °C overnight. During the cultivation period, the perceived turbidity in every tube was evaluated. The MIC is demonstrated as the lowermost concentration of the observed strain without turbulence.

### 3.8. Molecular Docking 

The possible mechanism of the biological process is recognized using molecular docking studies. The protein selected for perusing inhibitory action towards the microbial target is *E. coli* topoisomerase II DNA gyrase B (PDB ID: 1KZN) [49]. Molecular docking trials were conducted with Autodock Vina 1.1.2 to inspect the binding mechanism and inhibiting ability between CuO and Cu_2_O NPs, ciprofloxacin, and the 1KZN protein. The crystalline structure of the target protein was acquired from the Protein Data Bank (http://www.rcsb.org) for the purpose of conducting antibacterial examinations. The 3D structures of CuO and Cu_2_O nanoparticles and ciprofloxacin were created with the ChemBioOffice program package (v12.0). The program AutoDock Tools 1.5.6 was employed to produce the necessary files for docking. The search grid for the 1KZN protein was established using center coordinates (18.839, 26.702, 37.939) and dimensions (22, 20, 20), with a spacing of 1.0 Å. The exhaustiveness number was adjusted to 8, but all other limits for Vina docking were kept at their usual defaults. The substance with the smallest binding affinity score was selected as the highest-scoring substance, and the findings were examined with the Discovery Studio 2019 program.

## 4. Conclusions

The process of creating CuO and Cu_2_O NPs was accomplished by utilizing an extract from *Bidens pilosa* leaves and copper (II) chloride dihydrate in a water-based solution. The resulting nanoparticles were thoroughly examined, and their effectiveness towards drug-resistant bacterial strains was tested. The CuO and Cu_2_O NPs that were produced through biosynthesis were thoroughly analyzed utilizing several characterization methods. The SEM was utilized to characterize the morphological aspects of the biosynthesized CuO and Cu_2_O NPs. Based on these morphological evaluations, the biosynthesized CuO and Cu_2_O NPs were found to have sponge-like and spherical particles that were uniformly dispersed and aggregated. The XRD assessment was utilized to characterize the crystallinity and crystal structure of the produced nanoparticles. The XRD analysis exposed that the phytogenic synthesized CuO and Cu_2_O NPs exhibited a high degree of crystallinity and were determined to be of pure composition. The crystal size of the generated CuO and Cu_2_O NPs was estimated with the Scherrer equation from the XRD peaks. The crystal size was determined to be 12.45 nm for CuO and 68.19 nm for Cu_2_O, validating the synthesis of nanosized particles. The antibacterial characteristics of the produced CuO and Cu_2_O NPs towards various bacterial cultures were assessed by determining the MIC. The outcomes revealed that the nanoparticles displayed significant action towards the tested bacterial strains. The produced Cu_2_O NPs with a larger crystal size of 68.19 nm exhibited significant antibacterial properties towards all kinds of bacteria.

## Figures and Tables

**Figure 1 antibiotics-13-01088-f001:**
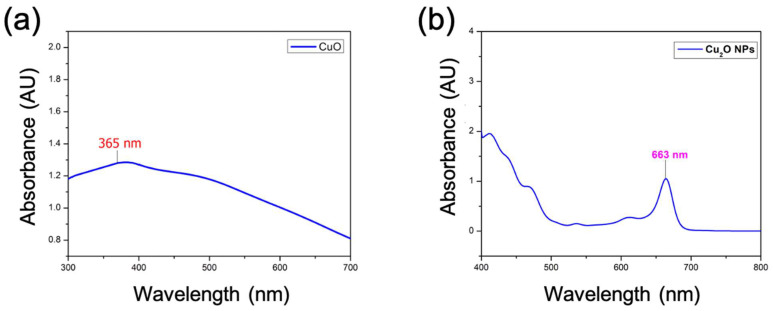
UV–vis spectra. (**a**) CuO with an absorption peak at 365 nm. (**b**) Cu_2_O nanoparticles (NPs) with an absorption peak at 663 nm.

**Figure 2 antibiotics-13-01088-f002:**
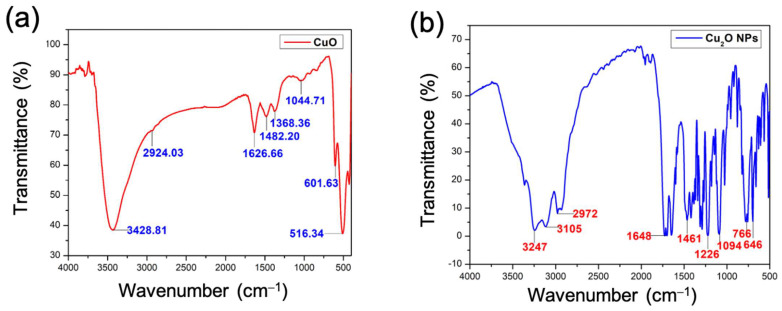
FT-IR spectra of the NPs: (**a**) CuO and (**b**) Cu_2_O.

**Figure 3 antibiotics-13-01088-f003:**
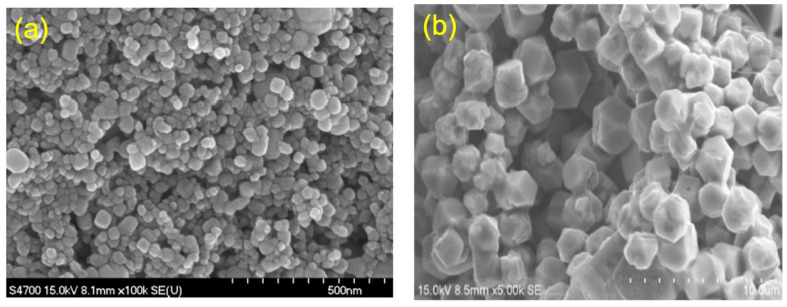
SEM image: (**a**) CuO and (**b**) Cu_2_O NPs.

**Figure 4 antibiotics-13-01088-f004:**
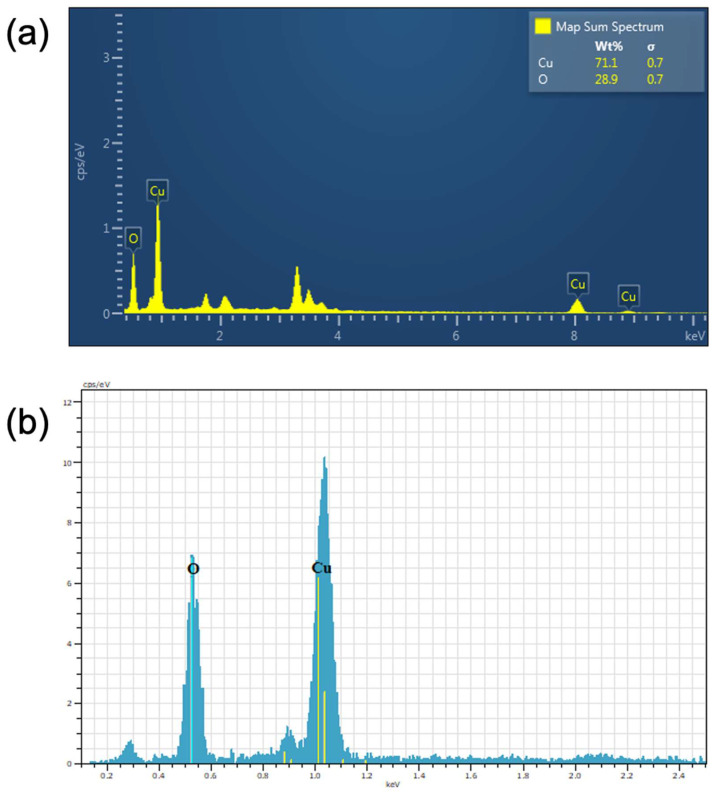
EDX spectra: (**a**) CuO and (**b**) Cu_2_O NPs.

**Figure 5 antibiotics-13-01088-f005:**
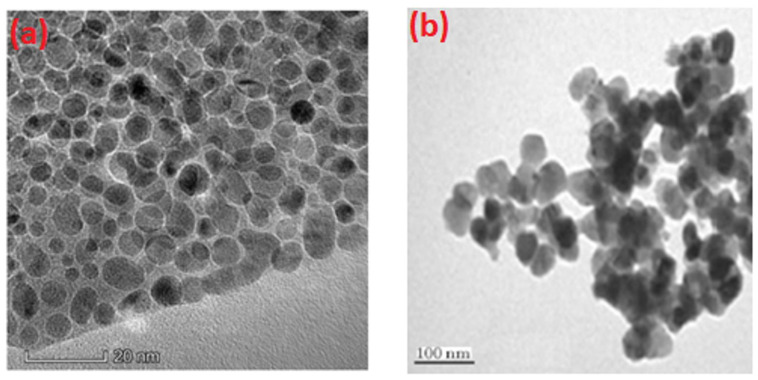
TEM image: (**a**) CuO and (**b**) Cu_2_O NPs.

**Figure 6 antibiotics-13-01088-f006:**
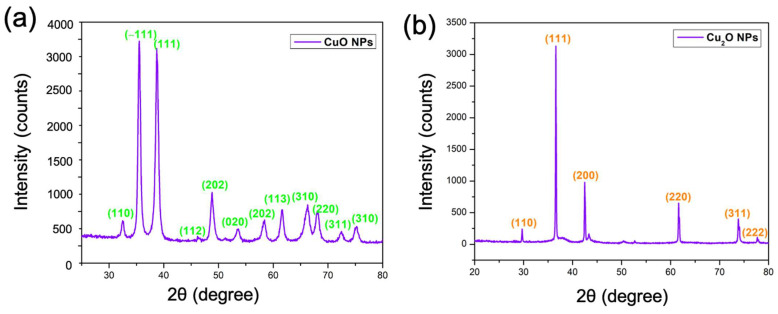
XRD spectra: (**a**) CuO and (**b**) Cu_2_O NPs.

**Figure 7 antibiotics-13-01088-f007:**
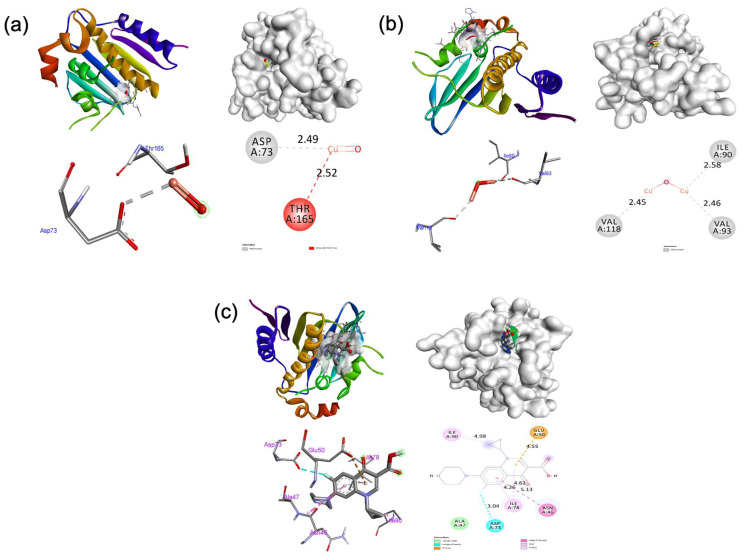
Interaction modes of synthesized (**a**) CuO and (**b**) Cu_2_O NPs and (**c**) ciprofloxacin within the binding cavity of the 1KZN receptor.

**Table 1 antibiotics-13-01088-t001:** Elemental makeup of CuO and Cu_2_O NPs.

NPs	Element	Atomic Number	Weight %	σ
CuO	O	8	28.90	0.7
	Cu	30	71.10	0.7
	Total	-	100	-
Cu_2_O	O	8	33.06	0.3
	Cu	30	66.94	0.3
	Total	-	100	-

**Table 2 antibiotics-13-01088-t002:** Antibacterial activity of CuO and Cu_2_O NPs.

Comp. No.	MIC µg/mL			
	*S. aureus*	*P. aeruginosa*	*E. coli*	*B. cereus*
*Bidens pilosa* Extract	28 ± 0.32	36 ± 0.64	30 ± 0.16	43 ± 1.34
CuO NPs	23 ± 0.64	30 ± 0.72	18 ± 1.54	26 ± 0.56
Cu_2_O NPs	20 ± 0.26	26 ± 1.82	16 ± 0.46	24 ± 1.34
Ciprofloxacin	26 ± 1.34	28 ± 0.0	20 ± 0.0	50 ± 0.35

**Table 3 antibiotics-13-01088-t003:** Docking behavior of synthesized CuO and Cu_2_O NPs and ciprofloxacin against protein 1KZN.

Compounds	Antibacterial Target Protein (PDB ID: 1KZN)		
	Binding Score (kcal/mol)	No. of H-Bonds	H-Bonding Residues
CuO NPs	−6.4	-	-
Cu_2_O NPs	−8.2	-	-
Ciprofloxacin	−6.0	-	-

## Data Availability

The data presented in this study are available upon request from the corresponding author.

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
