# Peer review of "Phytogenic Synthesis of Cuprous and Cupric Oxide Nanoparticles Using Black jack Leaf Extract: Antibacterial Effects and Their Computational Docking Insights"

_antibiotics, 2024, doi:10.3390/antibiotics13111088_

Round 1
Reviewer 1 Report
Comments and Suggestions for Authors
The study was synthesized using appropriate and green synthesis techniques for this journal. However, the authors must make some corrections, which I suggest below, before publication.
1- Explain the other peaks formed in the FT-IR spectrum.
2- In the FT-IR spectrum, groups belonging to organic compounds were found, but only Cu and O are mentioned in the EDX results. How is this possible?
3- In the SEM images, both materials appear quite large compared to the TEM results. Can you explain the reason for this?
4- In the XRD diffraction pattern, the statement "No contaminant peaks were detected." was used for CuO, but when Figure 6.a was examined, 2 (111) index peaks were identified. Could one of these peaks belong to Cu2O?
5- How did you prevent the reduction of copper during the synthesis?
6- The crystal sizes of both materials are quite different from each other. Please discuss the antibacterial effects without ignoring the particle size.
Author Response
Reviewer’s opinion: The study was synthesized using appropriate and green synthesis techniques for this journal. However, the authors must make some corrections, which I suggest below, before publication.
Response: We would appreciate the reviewer, whom provided a positive and constructive comment on our work. The reviewer’s comments and our responses as follows.
Comment. 1: Explain the other peaks formed in the FT-IR spectrum
Response: All the absorption peaks existing in the FT-IR spectra were discussed in detail as suggested by the reviewer.
Comment. 2: In the FT-IR spectrum, groups belonging to organic compounds were found, but only Cu and O are mentioned in the EDX results. How is this possible?
Response: Thank you for your valuable feedback. The presence of organic groups in the FT-IR spectrum, despite the EDX analysis showing only Cu and O, can be explained by the synthesis and characterization being conducted in a room atmosphere. This exposure likely introduced hydroxyl and carboxyl groups from ambient moisture and COâ‚‚. These groups appeared in the FT-IR results, as FT-IR is sensitive to surface-bonded functional groups, even from environmental exposure. Meanwhile, the purpose of the EDX analysis was to confirm the presence and approximate proportions of the primary elements, Cu and O, relevant to our study. We did not index additional peaks in the EDX data as they likely resulted from minor surface contamination and were not central to our analysis.
Comment.3: In the SEM images, both materials appear quite large compared to the TEM results. Can you explain the reason for this?
Response: The apparent size difference between the SEM and TEM images can be attributed to the distinct imaging principles and resolutions of these techniques. SEM typically provides surface topography and morphology at lower magnifications, which can make particles appear larger due to overlapping structures or surface aggregation. TEM, on the other hand, offers higher resolution and penetrative imaging, allowing for a clearer view of individual particles and finer structural details. As a result, TEM measurements often reflect the true nanoscale dimensions more accurately, whereas SEM may give the appearance of larger particle sizes due to surface effects and aggregation.
Comment.4: In the XRD diffraction pattern, the statement "No contaminant peaks were
identified. Could one of these peaks belong to Cu2O?
Response: Thank you for pointing this out, and we apologize for the oversight. The first peak in the XRD pattern indeed corresponds to the (-111) plane of CuO, not an additional (111) peak. We have corrected Figure 6a to reflect this accurately.
Comment. 5: How did you prevent the reduction of copper during the synthesis?
Response: Initially, the Cu2+ ions in the precursor was reduced to CuO by plant extract to form its metallic nanoparticles. Then the metallic nanoparticles were converted to metal oxide nanoparticles by heating them in hot air oven.
Comment. 6: The crystal sizes of both materials are quite different from each other. Please discuss the antibacterial effects without ignoring the particle size.
Response: Thank you for this insightful comment. We have discussed size and activity in the revised manuscript.
Reviewer 2 Report
Comments and Suggestions for Authors
All necessary corrections were indicated in the manuscript.

Author Response
Comment: All necessary corrections were indicated in the manuscript. (please find the attached manuscript along with reviewer-2’s comments)
Response: The necessary corrections were indicated in the manuscript have been made in the revised manuscript.
Reviewer 3 Report
Comments and Suggestions for Authors
Can be accepted after an editing correction, there were a few typos and some figure captions can be improved, some of the colors are too bright to see. Thanks
Author Response
Comment: Can be accepted after an editing correction, there were a few typos and some figure captions can be improved, some of the colors are too bright to see. Thanks
Response: Thank you for your valuable suggestion. The suggested additions have been made in the revised manuscript.